# Updates in the Role of Checkpoint Inhibitor Immunotherapy in Classical Hodgkin’s Lymphoma

**DOI:** 10.3390/cancers14122936

**Published:** 2022-06-14

**Authors:** Shazia Nakhoda, Farsha Rizwan, Aldana Vistarop, Reza Nejati

**Affiliations:** 1Department of Hematology/Oncology, Fox Chase Cancer Center, Philadelphia, PA 19111, USA; aldana.vistarop@fccc.edu (A.V.); reza.nejati@fccc.edu (R.N.); 2Department of Internal Medicine, Temple University Hospital, Philadelphia, PA 19140, USA; farsha.rizwan@tuhs.temple.edu

**Keywords:** checkpoint inhibitors, immunotherapy, Hodgkin lymphoma

## Abstract

**Simple Summary:**

The introduction of immunotherapy into the treatment options for Hodgkin’s lymphoma has improved survival in patients with recurrence of their cancer. These agents help the body’s immune system respond and clear cancer cells. Currently, these agents are only approved in patients who have had their cancer return multiple times or have progressed through multiple therapies. However, the evaluation of these agents as part of therapy in the upfront setting or second line setting has been conducted. This study will review these clinical studies and provide insights into the future directions of immunotherapy use by physicians in treatment of patients with Hodgkin’s lymphoma.

**Abstract:**

Classical Hodgkin’s lymphoma is a highly curable disease, but 10–25% of patients with higher-risk disease relapse. The introduction of checkpoint inhibitors (CPIs) targeting PD-1 have changed the landscape of treatment for patients with relapsed/refractory disease to multiple lines of therapy. The depth of response to CPI as a monotherapy is highest in the first relapse as salvage therapy based on outcomes reported in several phase II studies. With earlier use of CPI and brentuximab vedotin, the optimal sequencing of therapy is evolving. In this review, we will summarize clinical investigation of anti-PD-1 mAb in earlier line settings to provide insights on utilizing these agents as chemotherapy- and radiation-sparing approaches, increasing depth of response, and as part of combination regimens.

## 1. Introduction

Classical Hodgkin’s lymphoma (cHL) accounts for roughly 11% of all lymphomas in the United States [1] and is defined by its distinct pathologic features consistent with Reed Sternberg cells in a background of inflammatory cells, which enable the evasion of immune-mediated antitumor mechanisms [2,3,4]. Although the majority of patients with cHL can be cured with standard chemoradiation, patients who are unresponsive to initial treatment or who are not candidates for curative regimens have poor outcomes [5,6,7]. Immune checkpoint inhibitors (CPIs) enable the reconstitution of the immune response to target malignant cells, offering an alternative mechanism of action from traditional cytotoxic chemotherapy and have thus transformed the treatment landscape for relapsed/refractory (R/R) cHL.

Approximately 10–25% of patients with advanced or early-stage unfavorable risk disease have disease relapse [8,9]. The standard of care on relapse consists of salvage chemotherapy followed by autologous stem cell transplant (ASCT). This approach cures only about 50% of patients and is associated with significant risk for treatment-related toxicity and secondary malignancy [10,11]. Older patients and those with medical comorbidities who are not candidates for intensive therapy remain at a particularly high risk for relapse and inferior survival [12]. With modern frontline treatment regimens, patients over the age of 60 years had a ~15% worse 3-year progression-free survival (PFS) compared with younger patients [13].

Although the addition of the anti-CD30 monoclonal antibody drug conjugate brentuximab vedotin (BV) to the cHL treatment armamentarium has improved outcomes, the rate of relapse in patients treated with BV-containing frontline therapy is still 18% at 5 year follow-up [14]. In the high-risk R/R population who received BV as maintenance therapy after ASCT, 41% of patients have progression of their disease [15]. In the multiple-relapse setting, these responses are not durable, with only 38% of patients treated with BV maintaining their complete response (CR) at 5 year follow-up [16,17]. Immunotherapy with CPI treatment has demonstrated remarkable efficacy in patients with relapse after multiple lines of therapy and primary refractory disease and in patients who are not candidates for intensive salvage treatment (Table 1). This has led to the investigation of these agents in earlier lines of treatment to improve survival rates while also reducing treatment-related toxicity. Herein, we will review the clinical trial data on anti-PD1 mAb in the upfront and R/R settings over the past 5 years. We will also discuss considerations for sequencing, combination regimens, and the roles of chemotherapy- and/or radiation-sparing approaches.

## 2. Immune Checkpoints and the Tumor Microenvironment (TME) in cHL

The interaction of the programmed death receptor ligands 1 and 2 (PDL1/CD274 and PDL2/CD273) on tumor cells with PD-1 receptors on T cells leads to reduced T-cell activation and proliferation, thus enabling tumors to evade immune response (see Figure 1) [42]. The amplification of PDL1 and PDL2 on tumor cells, coded on chromosome 9p24.1, is mediated by PIM serine/threonine kinases on tumor cells via the constitutive activation of the NFkB and JAK-STAT signaling pathways [43].

Anti-PD-1 monoclonal antibody (mAb) therapies reconstitute the immune surveillance of malignant cells, enabling a rapid reduction in Reed-Sternberg cells and PDL1+ inflammatory cells as well as a durable PD-1 blockade that persists even after treatment cessation based on patient samples during and after treatment [44]. A rise in CD4+ T cell receptors, dendritic cells, and mature, highly differentiated NK cells is seen in peripheral blood after anti-PD1 mAb treatment, correlating with earlier and more robust clinical response [44,45,46,47].

PDL-1 expression is a well-defined marker of response to anti-PD-1 mAb across multiple cancer subtypes. In addition to this, 9p24.1 copy number amplification and the expression of major histocompatibility complex class II are associated with improved response to CPI in cHL [48]. In the frontline setting, these biomarkers did not demonstrate an association with early response rates with anti-PD-1 mAb-based therapy [49]. Emerging data on the role of metabolic disorders and patient demographic factors such as sex, age, and gender on effector T cell activity and PD-1 expression suggest that these factors may also contribute to CPI efficacy [50]. The further investigation of these tumoral biomarkers and the effects of extra-tumoral risk factors on TME may help guide treatment decision-making in the future.

Lastly, other immune checkpoints have been identified as potential targets in the TME of cHL, including cytotoxic T-lymphocyte-associated protein 4 (CTLA-4), lymphocyte-activation gene-3 (LAG-3), and T-cell immunoglobulin and mucin-domain containing 3 (TIM-3) [51,52]. The CD68 ligand that binds CTLA-4 is commonly expressed on HRS cells, and this expression persists after treatment with PD-1 blockade. This has therefore served as the rationale for targeting this protein with the monoclonal antibody ipilimumab to overcome anti-PD-1 mAb resistance [51]. Although LAG-3 and TIM-3 are not as frequently expressed on HRS cells, these proteins are commonly seen within the TME surrounding HRS cells [52]. However, ongoing investigation is needed to identify the clinical utility of targeting these alternative immune checkpoints. 

## 3. Checkpoint Inhibitors in Multiple Relapsed/Refractory Disease

For patients with R/R disease after multiple lines of therapy including ASCT +/− BV, anti-PD-1 mAb monotherapy has an overall response rate (ORR) of ~70% with a median PFS of 14–15 months. Although only 20–30% of responders have CR in this setting, those who do achieve CR have a prolonged remission with a median duration of response (mDOR) of 37 months with nivolumab; additionally, mDOR was not reached with pembrolizumab at 5 year follow-up [23,53]. In a study of anti-PD-1 mAb tislelizumab in patients with relapse after ASCT or 2 systemic regimens in a Chinese population, 67% of patients achieved CR, and ~50% of patients had ongoing response at 30 month follow-up, confirming the correlation of depth of response to duration of response with this class of agents [27]. Novel combination regimens of immunotherapy combined with other targeted therapies have deepened response rates in the multiple-relapse setting. The combinations of BV with the checkpoint inhibitors nivolumab and ipilimumab achieved CRs of 61% and 57%, respectively. Although the combination of all three agents achieved the highest CR of 73%, the inclusion of ipilimumab resulted in higher toxicity than BV + Nivo in this phase I/II study. The rates of grade 3–4 adverse events were reported at 43% and 50%, respectively, in the BV-Ipi and triplet combination arms but only 16% in the BV-Nivo group. [31]. Therefore, anti-PD-1 mAb based therapies are preferred.

In early clinical investigation, the addition of hypomethylating agent to anti-PD-1 mAb improved CR from 32 to 71% in patients with R/R cHL to at least 2 prior lines of therapy, with 100% of patients still responding to combination therapy at 6 month follow-up. Investigational regimens evaluating nivolumab in combination with JAK2 inhibitor, and BTK inhibitor are also underway (Table 2). Additionally, retreatment with anti PD-1 mAb has been described in a small number of patients with multiply relapsed disease, demonstrating ORRs of 70–100% and CRs of 33–73% [54,55,56,57]. Phase II investigation of combination BV + Nivo in patients with prior treatment of either agent is underway and may provide more insight into the roles of sequencing and retreatment as these agents are introduced to patients in earlier lines of therapy (Table 2).

## 4. Immunotherapy as First Salvage and Bridge to ASCT

Treatment with standard chemotherapy options in first relapse followed by ASCT leads to long-term cure in only half of patients, with reported 6 year event-free survival and overall survival (OS) of 45% and 55%, respectively [58]. Standard cytotoxic chemotherapy salvage achieved an ORR of ~70–90% and a CR of ~20–30% [59,60,61,62,63]. In patients who fail to respond to initial salvage chemotherapy, the outcomes after ASCT are particularly dismal. At 10 year follow-up, the PFS and OS are only 7–23% and 11–17%, respectively [60,61] Achieving complete metabolic response (CMR) to salvage therapy prior to ASCT is associated with superior outcomes and has served as a rationale for incorporating targeted therapies into first salvage lines of therapy. While the addition of BV into first-line salvage regimens has improved CR to 70–80% [62,64,65,66], single-agent BV induces CRs of only 27–43% when assessed as monotherapy or as part of PET-adapted bridging therapy ASCT [67,68].

On the other hand, the investigation of anti-PD-1 mAb in the first relapse in several phase II studies has demonstrated increased depth of response both as a single agent and as combination therapy (Table 1) [33,34,35]. Combination BV and Nivo demonstrated ORR of 82–85% and CR of 59–67% [32,69,70] At 3 year follow-up after combination achieved a PFS of 77%, and in patients who responded to induction and proceeded to ASCT, PFS improved to 91% [33]. In a small phase II PET-adapted study evaluating nivolumab monotherapy followed by escalation to nivolumab with ifosfamide, carboplatin, and etoposide (ICE) in patients with positive interim PET scans, 71% of patients achieved CR to the single agent nivolumab and were eligible to proceed directly to ASCT, sparing 26 of the total 42 evaluable patients from requiring cytotoxic treatments prior to ASCT. A total of 9 of the 42 patients required escalation of their treatment to N-ICE, and of these, 89% achieved CR [35]. This approach demonstrated overall 2 year PFS and OS of 72% and 95%, respectively (Table 1) [35]. Larger studies are underway to evaluate nivolumab monotherapy as salvage prior to ASCT (Table 2).

Other small studies evaluating combination of immunotherapy with standard regimens as bridge to ASCT, including pembrolizumab combined with ICE or gemcitabine, vinorelbine, and pegylated liposomal doxorubicin (GVD), showed similarly high CR rates of 86–95% [36]. The active investigation of other anti-PD-1 mAb combination regimens via a PET-adapted approach using anti-PD1 mAb with GVD are underway to assess efficacy with less toxic or chemotherapy-sparing therapies (Table 2).

## 5. Immunotherapy in Primary Refractory Disease

Historical data in patients with R/R disease within 1 year of frontline therapy who are treated with standard salvage chemotherapy followed by ASCT demonstrated a 10 year freedom from second failure (FF2F) and OS of 34 and 43%, respectively [9]. The incorporation of CPI has significantly improved outcomes in this population. In a subgroup analysis of R/R patients treated with pembrolizumab, patients with primary refractory disease achieved a CR of 35% with mDOR of 17 months at a median of 28 month follow-up, achieving 2 year PFS and OS of 32% and 94%, respectively. In a phase II study of pembrolizumab with chemotherapy (GVD) as initial salvage therapy as a bridge to ASCT, 79% of patients had either primary refractory disease or relapse within 1 year. All patients responded to this combination treatment with a remarkable CR of 95%, and 36 of the 39 patients enrolled proceeded to ASCT. All remained in remission at 13.5 month follow-up [34].

## 6. Immunotherapy as Maintenance after ASCT

Lastly, CPI has also been explored as consolidation therapy after ASCT similar to BV, as investigated in the phase III ATHERA study [15,37]. In the ATHERA study, high-risk patients, defined as primary refractory or relapse/progression of disease within 12 months of frontline therapy or extra-nodal involvement at relapse, were included. Extended follow-up of BV maintenance demonstrated a 5-year PFS of 59% vs 41% with placebo [37]. Phase II evaluation of pembrolizumab after ASCT demonstrated an estimated 18 month PFS of 82% overall, and among patients in the subgroup who met the criteria for high-risk disease per the ATHERA protocol, PFS was 85% [15,37].

In the absence of long-term follow-up and head-to-head comparison, the PFS with post-ASCT consolidative pembrolizumab compares favorably with that of BV. However, as BV has moved to the upfront setting as the standard of care for advanced-stage cHL and nivolumab is being used in earlier lines of therapy, the role of maintenance BV or anti-PD-1 mAb is less clear. Ongoing phase II investigation of the combination of BV and nivolumab in patients with relapse who previously received either agent may provide insight into the roles of sequencing and retreatment with these agents as consolidation post ASCT (Table 2).

## 7. Immunotherapy in Frontline Treatment

Frontline chemotherapy with doxorubicin, vinblastine, and dacarbazine with bleomycin (ABVD) or BV (A-AVD) with and without radiation (RT) achieve high cure rates. However, these therapies are associated with significant toxicity such as bleomycin- and/or radiation-associated lung toxicity, anthracycline-associated cardiotoxicity, infections, and secondary malignancy [7,11,71,72] Whereas the reduction or elimination of bleomycin as part of a PET-adapted ABVD regimen or the A-AVD regimen has reduced rates of pulmonary toxicity and the use of G-CSF has decreased the risk of infectious complications [13,73,74] older and medically frail patients remain at high risk for complications [72,75,76]. An analysis of patients treated with ABVD from the German Hodgkin Study Group demonstrated that 68% of older patients suffered from major toxicities, with a treatment-related mortality of 5% [77].

Although anti-PD-1 mAb is associated with unique immune-related adverse events (irAEs), the majority are mild and can be well managed with observation or immunosuppression based on severity. Grade 3–4 irAE with anti-PD-1 mAb is reported at rate of 12–28% in patients with cHL which is similar to rates described in patients with solid tumors [22,40,78] In a meta-analysis of patients with cancer treated with anti-PD-1 mAb, the rate of treatment-related mortality was 0.36% [79]. This has therefore led to investigation of anti-PD-1 mAb in the frontline setting.

Unlike in the second-line salvage setting, single-agent anti-PD-1 mAb and mAb in combination with BV failed to improve response rates in the frontline setting compared with standard therapy [39,41]. However, 37–51% of patients did achieve CR to single-agent PDL-1 mAb when given nivolumab or pembrolizumab initially as part of sequential therapy followed by AVD, identifying a population that may be spared from cytotoxic treatment [39,41]. In the NIVAHL study, patients received N-AVD as concomitant therapy or sequential treatment with nivolumab followed by AVD, both as part of combined modality treatment for early-stage unfavorable disease. Combination N-AVD induced a CR of 87% compared with 51% after the initial course of nivolumab. Although this study failed to meet its efficacy benchmark in the concomitant cohort, the 1 year PFS rates were 100% and 98% in the concomitant and sequential therapy arms, respectively [39].

In a study evaluating pembrolizumab with AVD as a radiation-sparing (RT) approach in early unfavorable or advanced-stage disease, all patients received sequential therapy. Specifically, 37% had CR to initial pembrolizumab monotherapy, and all patients achieved CR after 2 additional cycles of AVD. At the median follow-up of 22 months, all patients had sustained response, and none were treated with RT. Of the 30 total patients, 12 had bulky disease [41].

In a study of upfront BV and nivolumab in older patients and those unsuitable for standard chemotherapy, only 64% of patients responded, and this study was closed early due to failure to meet the primary endpoint [38]. At the median follow-up of 21 months, the mPFS was 22 months, and mDOR was not reached. Although this study did not improve standard of care therapy, it demonstrates high activity as a chemotherapy-sparing approach in those who are not candidates for more intensive therapy. Based on the remarkable efficacy of anti-PD-1 mAb therapy in chemotherapy-refractory disease, treatment with immunotherapy in the frontline as part of a PET-adapted approach in patients with suboptimal response to anthracycline-based treatment is under investigation (Table 1). In early-stage non-bulky disease, PET-adapted treatment with nivolumab-based therapy as an RT-sparing option or in combination with RT is also underway (Table 2).

## 8. Future Directions

The optimal sequencing of CPI, brentuximab vedotin, cytotoxic therapy, and ASCT in the R/R setting remains unclear as targeted agents are now being used in the upfront and 2nd-line setting. Therefore, the future treatment paradigm of cHL will continue to evolve based on the multiple active clinical trials summarized in Table 2, particularly the comparison of upfront N-AVD vs. A-AVD in advanced-stage disease (NCT03907488). Determining which patients are eligible for retreatment with anti-PD-1 mAb is crucial and will likely be dependent on the duration of the initial response and the reason for discontinuation. Novel anti-PD-1-based combination regimens with a hypomethylating agent, a Bruton’s tyrosine kinase inhibitor, and a JAK2 inhibitor may be effective in heavily pretreated patients and/or those with prior progression on CPI regimens.

In future trial designs for the treatment of cHL in the upfront setting, efforts to reduce short- and long-term toxicity rates must be prioritized in addition to malignancy-specific survival metrics. Although small phase II studies have demonstrated favorable efficacy of anti-PD-1 mAb with chemotherapy as a RT-sparing approach in early-stage bulky disease, ongoing studies using a PET-adapted approach are underway. By incorporating anti-PD-1 mAb as treatment escalation in patients with positive interim PET findings after upfront chemotherapy, patients have the potential of avoiding additional toxic chemotherapy and radiation (NCT04866654, NCT03233347, NCT03712202). 

## 9. Conclusions

Classic HL is characterized by the dysregulation of TME-enabling pro-tumoral signaling and the evasion of immune surveillance by cancer cells, for which immune checkpoint inhibitor therapy is an ideal target. The inclusion of anti-PD-L1 mAb into the treatment armamentarium of cHL has provided patients with R/R disease who are insensitive to cytotoxic chemotherapy with the potential for long-term remission, particularly in those achieving deep response to treatment.

Single-agent anti-PD-1 mAb offers a CR of ~20–30% in the multiply relapsed/refractory setting, ~60–70% in the first relapse setting, and ~40–50% in the frontline setting based on various phase II studies (Table 1). It can be concluded that although standard chemoradiation can cure most patients in the upfront setting, those who do relapse have disease biology driven primarily by immune dysregulation and insensitivity to cytotoxic chemotherapy. This therefore serves as the ideal rationale for PET-adapted approaches in the frontline setting where patients who fail to have CMR on PET imaging with standard cytotoxic therapy proceed to anti-PD-1 mAb-based treatment (Table 2). It also serves as rationale for the utilization of immunotherapy in the 2nd-line salvage setting prior to ASCT as anti-PD-1 mAb utilizes an intact immune system that may be compromised with repeated lines of cytotoxic chemotherapy.

The specific populations that have the greatest potential for benefit from anti-PD-1 mAb-based therapy are patients deemed poor candidates for standard chemotherapy due to age or medical comorbidities and those with refractory or early relapsing disease. Based on the current data, a decision to incorporate immunotherapy must be tailored to individual patients based on their ASCT eligibility, their underlying comorbidities including those that put them at increased risk for immune-mediated toxicities related to anti-PD-1 mAb, and their personalized goals of care.

## Figures and Tables

**Figure 1 cancers-14-02936-f001:**
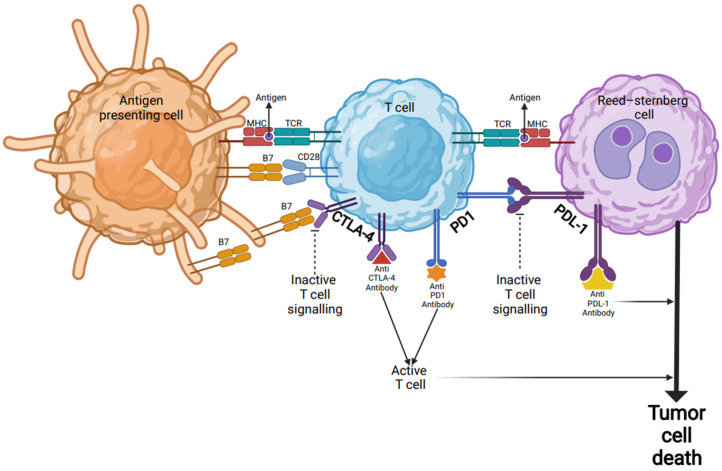
Checkpoint inhibitor immunotherapy in Hodgkin’s lymphoma. Inactive T cells are activated through their TCR by encountering antigenic peptides presented by the MHC complex on the surfaces of APCs or tumor cells (Reed-Sternberg cells). In addition to TCR-MHC engagement, a co-stimulatory signal via B7 protein is required for target-cell lysis and effector cell responses. B7 protein on activated APCs can pair with either a CD28 on the surface of a T-cell to produce a costimulatory signal to enhance the activity of TCR-MHC signal and T-cell activation, or it can pair with CTLA-4 to produce an inhibitory signal to keep the T cell in the inactive state. Blocking the binding of B7 to CTLA-4 with an anti-CTLA-4 antibody allows T cells to be activated and to kill tumor cells. Other immune checkpoint proteins such as PD-1 on the surfaces of T cells and PD-L1 on tumor cells can also prevent T cells from killing tumor cells. Immune checkpoint blockade via monoclonal antibodies (anti-PD-L1 or anti-PD-1) can lead T cells to kill tumor cells. Abbreviations: MHC: major histocompatibility complex; APC: antigen-presenting cells; TCR: T-cell receptor; CTLA-4: cytotoxic T-lymphocyte associated antigen 4; PD-1: programmed death 1; PD-L1: programmed death-ligand 1.

**Table 1 cancers-14-02936-t001:** Phase II and III Studies Evaluating Anti-PD1 mAb Therapies in Hodgkin’s Lymphoma.

Study	Phase	Key Inclusion	Agent	*n*	ORR (CR)	PFS (Median Follow Up)	OS	mDOR (Median Follow Up)
Multiple R/R Disease: anti-PD-1 mAb monotherapy								
Younes 2016 [18] Armand 2018 [19] Ansell 2021 [20]	II	Post ASCT +/− BV	Nivo	243	71% (21%)	37% (24 m) 18% (60 m)	87% (24 m) 71% (60 m)	18 m (58 m)
Chen 2017 [21] Chen 2019 [22] Armand 2021 [23]	II	Post ASCT +/− BV or R/R to first line salvage therapy	Pembro	210	71% (28%)	44% (60 m)	71% (60 m)	7 m (60 m)
Zinzani 2020 [24]	II	Primary Refractory Disease subgroup	Pembro	71	82% (35%)	32% (24 m)	94% (24 m)	17 m (28 m)
Kuruvilla 2021 [25]	III	Post or ineligible for ASCT	Pembro vs. BV	304	N/A	54% vs. 36 (12 m)	N/A	N/A
Song 2020 [26] Song 2022 [27]	II	Post or ineligible for ASCT	Tislelizumab	70	87% (67%)	41% (36 m)	85% (36 m)	32 m (10 m)
Nie 2019 [28] Liu 2021 [29]	II	Post 2+ prior LOT	Camrelizumab	19	90% (32%)	67% (24 m)	63% (24 m)	NR
Song 2019 [30]	II	Post ASCT	Camrelizumab	75	76.0% (28%)	81% (6 m) 67% (12 m)	NR	NR
Multiple R/R Disease: anti-PD-1 mAb combination therapies								
Diefenbach 2020 [31]	I/II	R/R after 1+ prior LOT	Ipi/BV	21	76% (57%)	61% (12 m)	NR	N/A
			Nivo/BV	18	89% (61%)	70% (12 m)	NR	NR
			Ipi/Nivo/BV	22	82% (73%)	80% (12 m)	NR	NR
Lepik 2020 [32]	II	R/R to Nivo monotherapy	Nivo + Benda	30	87% (57%)	23% (25 m)	97% (24 m)	7 m (25 m)
Nie 2019 [28] Liu 2021 [29]	II	R/R in anti-PD-1 mAb naïve pts	Camrelizumab + Decitabine	42	95% (71%)	79% (6 m) 89% (12 m)	63% (24 m)	NR
		R/R in anti-PD-1 resistant pts	Camrelizumab + Decitabine	25	52% (28%)	79% (6 m) 59% (12 m)	N/A	16 m (35 m)
First Salvage prior to Transplant								
Advani 2021 [33]	I/II	R/R in first salvage therapy	Nivo+ BV	91	85% (67%)	77% (36 m)	93% (36 m)	N/A
Moskowitz 2021 [34]	II	R/R prior to ASCT	Pembro + GVD	38	100% (95%)	N/A	N/A	N/A
Mei 2022 [35]	II	R/R bridge to ASCT	Nivo +/− ICE	9	100% (89%)	72% (24 m)	94% (24 m)	N/A
Bryan 2021 [36]	II	R/R prior to ASCT	Pembro + ICE	42	N/A	88% (24 m)	95% (24 m)	N/A
Maintenance after ASCT								
Armand 2019 [37]	II	R/R after ASCT	Pembro	30	N/A	82% (18 m)	100% (18 m)	N/A
Frontline								
Cheson 2020 [38]	II	60+ years old and ineligible for chemotherapy	Nivo + BV	46	61% (48%)	N/A	N/A	N/A
Brockelmann 2020 [39]	II	Early stage unfavorable	Nivo + AVD (C and S)	109	96% (87%)	S: 95% (24 m) C: 100% (24 m)	S: 100% (24 m) C: 100% (24 m)	N/A
Ramchandren 2019 [40]	II	Advanced stage	Nivo + AVD (S)	51	84% (67%)	92% (9 m)	98% (9 m)	N/A
Allen 2021 [41]	II	Early stage unfavorable and advanced stage	Pembro + AVD (S)	30	100% (No CR)	NR	NR	N/A

AVD: Adriamycin, vinblastine, dacabazine; C: combination; CR: complete response rate assessed by positron emission tomography (PET); GVD: gemcitabine, vinorelbine, doxorubicin (liposomal); ICE: ifosfamide, carboplatin, etoposide; Ipi: ipilimumab; LOT: line of therapy; mDOR: median duration of response; N/A: data not available; NR: not reached; Nivo: nivolumab; ORR: overall response rate assessed by PET; PFS: progression-free survival; Pembro: pembrolizumab; S: sequential; m: month.

**Table 2 cancers-14-02936-t002:** Phase II/III Pipeline Studies Evaluating Anti-PD1 mAb Therapies in Hodgkin’s Lymphoma.

ClinicalTrial.Gov Identifier	Phase	Regimen
Frontline Setting Regimens		
NCT04866654	II	ABVD +/− PET adapted RT + Nivo in early stage nonbulky disease
NCT03233347	II	A-AVD with PET adapted BV + Nivo; followed by Nivo maintenance in early-stage nonbulky disease
NCT03712202	II	ABVD with PET adapted BV/Nivo or ABVD in PET negative interim scan; A-AVD+ Nivo in PET positive interim scan in early-stage disease
NCT03907488	III	A-AVD vs. Nivo-AVD in stage III/IV disease
NCT03033914	I/II	A(B)VD +/− PET adapted Nivo-AVD in stage III/IV disease
NCT03580408	II	Nivo +/− Vinblastine in patients aged > 60 years and unfit for standard chemotherapy in any stage disease
Relapsed/Refractory setting: Combination with Chemotherapy Regimens		
NCT03739619	I/II	Nivo−Bendamustine + Gemcitabine
NCT04091490	II	Nivo−DHAP
Relapsed/Refractory setting: Combination with Radiation Regimens		
NCT03480334	II	Nivo + RT in R/R cHL
NCT03495713	II	Nivo + low dose RT in R/R cHL
Relapsed/Refractory setting: Chemotherapy Sparing Regimens		
NCT04624984	II	PD-1 inhibitor vs. PD-1 inhibitor + GVD as first salvage
NCT03337919	II	Nivo monotherapy in disease refractory to 1st/2nd line salvage prior to ASCT
NCT04938232	II	Ipi +/− Nivo in R/R cHL
NCT04561206	II	BV + Nivo as 2nd line tx in patients not candidates for ASCT
Relapsed/Refractory setting: Combination with Other Targeted Therapy Regimens		
NCT01896999	I/II	Nivo + BV +/− Ipi in R/R cHL
NCT05137886	II	Tislelizumab + Decitabine
NCT03681561	I/II	Nivo + Ruxolitinib in R/R cHL
NCT02940301	II	Nivo + Ibrutinib in R/R cHL
NCT03057795	II	BV + Nivo consolidation post ASCT
NCT05039073	II	BV + Nivo combination in patients previously treated with BV or CPI

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
