# Peer review of "Updates in the Role of Checkpoint Inhibitor Immunotherapy in Classical Hodgkin’s Lymphoma"

_cancers, 2022, doi:10.3390/cancers14122936_

Round 1
Reviewer 1 Report
In present review, authors present the overview of immunotherapy treatment to the patients with classical Hodgkin Lymphoma. I have several reservations, my comments are appended as below:
- Introduction- reference 5-7- please indicate HR, P value.
- Immunotherapy is known to affected by additional cofounders like obesity, smoking history. Author’s should note and discuss PMID: 33076303.
- Response to immunotherapy is dictated by receptors other than PD-L1. Do authors note that?
- How CR/R/PFS were defined?
- Immunotherapy and toxicity- elaborate (section-Immunotherapy in Frontline Treatment).
- Effect of Checkpoint Therapy on Tumor Microenvironment in cHL: are there reports in young vs older patients?
- There should be ‘future directions’ sections.
Author Response
Reviewer #1’s Comments:
- Introduction- reference 5-7- please indicate HR, P value.
- Response: Thank you for this thoughtful suggestion. Upon our further consideration of the points highlighted in this section, it seems discussion of these historical clinical trials (reference 5-7) are 1) redundant (as prognosis is further described in the 2nd paragraph) and 2) out of scope for this specific topic. Therefore, there is little value of adding more to the introduction describing these studies and their statistical significance. These references have been removed from this section and we have changed the sentence to simply read “While the majority of patients with cHL can be cured with standard chemoradiation, patients who are unresponsive to initial treatment or not candidates for curative regimens have poor outcomes.”
- Response: Thank you for this thoughtful suggestion. Upon our further consideration of the points highlighted in this section, it seems discussion of these historical clinical trials (reference 5-7) are 1) redundant (as prognosis is further described in the 2nd paragraph) and 2) out of scope for this specific topic. Therefore, there is little value of adding more to the introduction describing these studies and their statistical significance. These references have been removed from this section and we have changed the sentence to simply read “While the majority of patients with cHL can be cured with standard chemoradiation, patients who are unresponsive to initial treatment or not candidates for curative regimens have poor outcomes.”
- Immunotherapy is known to affected by additional cofounders like obesity, smoking history. Author’s should note and discuss PMID: 33076303.
- Response: Thank you for this suggestion and providing us with this interesting reference. We have addressed this in Section 2 titled “Immune Checkpoints and the Tumor Microenvironment (TME) in cHL.”
- Response: Thank you for this suggestion and providing us with this interesting reference. We have addressed this in Section 2 titled “Immune Checkpoints and the Tumor Microenvironment (TME) in cHL.”
- Response to immunotherapy is dictated by receptors other than PD-L1. Do authors note that?
- Response: Thank you for this suggestion. We have included data on other biomarkers for response to anti-PD-1 mAb in the 2nd paragraph of Section 2 titled “Immune Checkpoints and the Tumor Microenvironment (TME) in cHL.” We also included additional information on other immune checkpoints that serve as targets for therapy in CHL such as CTLA-4, LAG-3, and TIM-3. This has been summarized in an additional third paragraph of the same section.
- How CR/R/PFS were defined?
- Response: Thank you for this comment. We have added the phrase “assessed on positron emission tomography (PET) in the caption describing Table 1.
- Response: Thank you for this comment. We have added the phrase “assessed on positron emission tomography (PET) in the caption describing Table 1.
- Immunotherapy and toxicity- elaborate (section-Immunotherapy in Frontline Treatment).
- Response: Thank you for this recommendation. We have added further information on toxicity in the Section 7: Immunotherapy in Frontline Treatment.
- Response: Thank you for this recommendation. We have added further information on toxicity in the Section 7: Immunotherapy in Frontline Treatment.
- Effect of Checkpoint Therapy on Tumor Microenvironment in cHL: are there reports in young vs older patients?
- Response: Most of these reports were in younger patients as these were patients included on clinical trials and were candidates for stem cell transplant. Despite the likely significance of age and its impact on immune response, the effect of age is not specifically addressed in these reports. Therefore, we have not included this comment in the manuscript.
- Response: Most of these reports were in younger patients as these were patients included on clinical trials and were candidates for stem cell transplant. Despite the likely significance of age and its impact on immune response, the effect of age is not specifically addressed in these reports. Therefore, we have not included this comment in the manuscript.
- There should be ‘future directions’ sections
- Response: We appreciate this recommendation. We have added a section titled “Future Directions” which has now incorporated two paragraphs from the conclusion that touched on this topic in the initial submission.
Reviewer 2 Report
This paper is a very nice and well-written article about immunotherapy in classical Hodgkin lymphoma. Non additional modifications/revisions are required.
Author Response
Thank you kindly for reviewing our submission.
Reviewer 3 Report
Major Points of Criticism:
(1) In their title the authors promise an update on „immunotherapy“ in classical Hodgkin lymphoma. However, they concentrate mainly on checkpoint inhibitor therapy and do not provide a general overview on immunotherapy for Hodgkin lymphoma per se which would include also for example more details on anti-CD30 monoclonal antibodies and also on anti-CD20 Rituximab. Hence the title should be changed and be more specific.
(2) The manuscript is very simply constructed, mainly very dry text and two endless tables listing various trials. There is not a single illustration or figure which would loosen up this very uninspired presentation. Even the simplest solution of showing in a figure the mechanisms of checkpoint inhibitior therapy was not taken up.
Minor and Specific Points of Criticism:
(1) Page 1, paragraph 1, line 3: The abbreviation „RS“ must be defined -> „Reedberg Sternberg (RS) cells“.
(2) Page 2, line 7: „Brentuximab“ should be defined as an anti-CD30 monoclonal antibody.
(3) Page 2, line 9: „R/R“ must be defined here at its first appearance (and not on page 4).
(4) Table 1 Headline: „ORR“ and „mDOR“ must be defined not later on page 4 but here when they occur first.
(5) Table 1 (and also in the text): Suffice to indicate the results in full percentages (without decimals which suggests only a false sense of accuracy). Also to indicate months with decimals is clearly over the top.
(6) Table 1: It would be useful to define the acronyms ICE, AVD, GVD already in the legend of Table 1 where they appear first (pages 2-4) and not significantly later downstream on later pages.
(7) Table and Table 2, legends: The abbreviations should be listed in alphabetical order (otherwise one has to screen the whole list in order to find a single acronym which possibly comes then at the end).
(8) Page 4, paragraph 2, line 1: The CD numbers could be added as follows: "(PDL1/CD274 and PDL2/CD273)".
(9) Page 4, paragraph 3: Abbreviating months as "m" is not necessary; so much space or time is not saved when writing "months" instead of "m" (also years is elsewhere written as word).
(10) Table 2, legend: The definition of the acronym PET belongs into the legend of Table 2 (and not later downstream.
(11) An Outlook („Future“) would be useful.
(12) The manuscript should be checked for many minor mistakes and typing errors, for example:
- „A rise in in“ instead of „A rise in“
- „comined“ instead of „combined (page 6),
- „proceed“ instead of „proceeded“ (page 7),
- „based their“ instead of „based on their“ (page 8),
- „“underly“ instead of „underlying“ (page 8)
– and many others.
Author Response
Reviewer #3’s Comments:
- In their title the authors promise an update on „immunotherapy“ in classical Hodgkin lymphoma. However, they concentrate mainly on checkpoint inhibitor therapy and do not provide a general overview on immunotherapy for Hodgkin lymphoma per se which would include also for example more details on anti-CD30 monoclonal antibodies and also on anti-CD20 Rituximab. Hence the title should be changed and be more specific.
- Response: Thank you for this thoughtful suggestion. We have changed the title to “ Updates in the Role of Checkpoint Inhibitor Immunotherapy in classical Hodgkin Lymphoma.” We have also included additional information about anti-CLTA-4 mAb, ipilimumab in the 2nd paragraph of section 3 titled “Checkpoint Inhibitors in Multiple Relapsed/Refractory Disease.” We also have described other immune checkpoints that serve as potential clinical targets in the 3rd paragraph of section 2 titled “Immune Checkpoints and the Tumor Microenvironment (TME) in cHL.”
- Response: Thank you for this thoughtful suggestion. We have changed the title to “ Updates in the Role of Checkpoint Inhibitor Immunotherapy in classical Hodgkin Lymphoma.” We have also included additional information about anti-CLTA-4 mAb, ipilimumab in the 2nd paragraph of section 3 titled “Checkpoint Inhibitors in Multiple Relapsed/Refractory Disease.” We also have described other immune checkpoints that serve as potential clinical targets in the 3rd paragraph of section 2 titled “Immune Checkpoints and the Tumor Microenvironment (TME) in cHL.”
- The manuscript is very simply constructed, mainly very dry text and two endless tables listing various trials. There is not a single illustration or figure which would loosen up this very uninspired presentation. Even the simplest solution of showing in a figure the mechanisms of checkpoint inhibitor therapy was not taken up.
- Response: We appreciate this recommendation to add a figure describing the mechanism of checkpoint inhibition which we have incorporated.
- Response: We appreciate this recommendation to add a figure describing the mechanism of checkpoint inhibition which we have incorporated.
- Minor and Specific Points of Criticism:
- Response: Thank you for noting these typing errors and suggesting clarification on abbreviations and technical terms. We have incorporated the outlined changes requested:
- Page 1, paragraph 1, line 3: The abbreviation „RS“ must be defined -> „Reed Sternberg (RS) cells“
- Page 2, line 7: „Brentuximab“ should be defined as an anti-CD30 monoclonal antibody.
- Page 2, line 9: „R/R“ must be defined here at its first appearance (and not on page 4).
- Table 1 Headline: „ORR“ and „mDOR“ must be defined not later on page 4 but here when they occur first.
- Table 1 (and also in the text): Suffice to indicate the results in full percentages (without decimals which suggests only a false sense of accuracy). Also to indicate months with decimals is clearly over the top.-
- Table 1: It would be useful to define the acronyms ICE, AVD, GVD already in the legend of Table 1 where they appear first (pages 2-4) and not significantly later downstream on later pages.
- Table and Table 2, legends: The abbreviations should be listed in alphabetical order (otherwise one has to screen the whole list in order to find a single acronym which possibly comes then at the end).
- Page 4, paragraph 2, line 1: The CD numbers could be added as follows: "(PDL1/CD274 and PDL2/CD273)".
- Page 4, paragraph 3: Abbreviating months as "m" is not necessary; so much space or time is not saved when writing "months" instead of "m" (also years is elsewhere written as word).
- Table 2, legend: The definition of the acronym PET belongs into the legend of Table 2 (and not later downstream.)
- The manuscript should be checked for many minor mistakes and typing errors, for example:
- „A rise in in“ instead of „A rise in“
- „comined“ instead of „combined (page 6),
- „proceed“ instead of „proceeded“ (page 7),
- „based their“ instead of „based on their“ (page 8),
- „“underly“ instead of „underlying“ (page 8)
- and many others.
- An Outlook („Future“) would be useful.
- Response: We appreciate this recommendation. We have added a section titled “Future Directions.”
- Response: Thank you for noting these typing errors and suggesting clarification on abbreviations and technical terms. We have incorporated the outlined changes requested:
Round 2
Reviewer 1 Report
All my comments are answered.